# Colorectal Cancer Presentation and Survival in Young Individuals: A Retrospective Cohort Study

**DOI:** 10.3390/cancers10120472

**Published:** 2018-11-28

**Authors:** Mark B. Ulanja, Bryce D. Beutler, Mohit Rishi, Chioma Ogala, Darryll R. Patterson, Nageshwara Gullapalli, Santhosh Ambika

**Affiliations:** 1Department of Internal Medicine, Reno School of Medicine, University of Nevada, Reno, NV 89557, USA; bbeutler@med.unr.edu (B.D.B.); mrishi@med.unr.edu (M.R.); cogala@med.unr.edu (C.O.); drpatterson@med.unr.edu (D.R.P.); ngullapalli@med.unr.edu (N.G.); Sambika@renown.org (S.A.); 2Renown Institute for Cancer, 1155 Mill Street, W-11, Reno, NV 89502, USA

**Keywords:** colorectal cancer, survival, metastatic, surveillance epidemiology and end results, young, cohort

## Abstract

Emerging evidence suggests that the incidence of colorectal cancer is increasing among individuals under the age of 50 years. However, the pattern of disease presentation in young patients remains under investigation. This is a retrospective cohort study of patients diagnosed with colorectal cancer (CRC) between 2004 and 2015. Data was acquired from the Surveillance, Epidemiology, and End Results 18 program registries. A total of 269,398 patients who met the inclusion criteria were included in the final analysis. The primary outcomes were the likelihood of metastatic disease at diagnosis and survival. Of the 269,389 patients diagnosed with CRC, 11.8% of the patients were young (20 to 49 years), 45.6% were middle-aged (50 to 69 years), and 42.6% were elderly (70 years or older). Individuals in the middle-aged and elderly cohorts were significantly less likely to present with metastatic disease as compared to the young cohort (middle-aged adjusted odds ratio (aOR) = 0.73, 95% confidence interval (CI) = 0.70 to 0.75, elderly aOR = 0.49, 95% CI = 0.47 to 0.50). However, overall survival was longest in the young cohort. We conclude that young individuals with colorectal cancer have an increased risk of presenting with distant metastases as compared to the middle-aged and elderly, but, nevertheless, exhibit prolonged survival.

## 1. Introduction

Colorectal cancer represents the third most common malignancy in the United States and emerging evidence suggests that the incidence of colorectal cancer (CRC) is increasing among individuals under the age of 50 years. However, the pattern of disease presentation in young patients remains under investigation.

The incidence of CRC in young individuals has increased by 2% to 8% annually over the past two decades [1,2]. CRC is now one of the 10 most common causes of death among individuals between the ages of 20 and 49 years [3]. Adults aged 50 years and older have had a decline in CRC. This has been attributed primarily to widespread implementation of screening guidelines [4]. Several retrospective studies have shown that younger patients have an increased risk of presenting with advanced stage disease as compared to older adults [5]. However, some investigators have reported contradictory results showing that younger patients are more likely to present with less-advanced stages of CRC [6,7]. Furthermore, the relationship between patient age at CRC diagnosis and survival remains under investigation. We, therefore, evaluated the National Cancer Institute’s Surveillance, Epidemiology, and End Results (SEER) 18 registries to determine if young patients with CRC are more likely to present with metastatic disease and to ascertain survival among this cohort. Age was grouped into 20–49 years (young), 50–69 years (middle age), and 70 years or older.

## 2. Results

### 2.1. Study Population

A total of 269,398 patients diagnosed with CRC are included in the final analysis. In addition, 129,538 (48.0%) were female and 139,860 (51.9%) were male. Overall, 11.8% of the patients were young (20–49 years), 45.6% were middle-aged (50–69 years), and 42.6% were elderly (70 years or older). Patients were further stratified into two groups based on year of diagnosis: EraA (2004–2009) and EraB (2010–2015). A significantly higher proportion of young patients were diagnosed in EraB as compared to EraA (*p* < 0.05). In addition, EraB had a significantly higher proportion of middle-aged (47.9%), female (52.0%), white (66.6%), and insured (81.6%-excluding Medicaid) patients.

### 2.2. Patient Characteristics

Baseline demographic, tumor, clinical, and treatment characteristics are presented in Table 1. Furthermore, 111,935 (41.6%) patients died over the study period in which 75,398 (67.4%) were in EraA and 36,537 (32.6%) were in EraB (*p* < 0.001). A total of 48,559 (18.0%) patients had metastatic CRC at diagnosis. Among these, 16.0% were young, 48.9% were middle-aged, and 35.1% were elderly. In contrast, a diagnosis of either stage 1, 2, or 3 CRC was established in 10.8% of young, 44.9% of middle-aged, and 44.3% of elderly individuals (Appendix A).

### 2.3. Likelihood of Metastatic Disease

In a logistic multivariable model for demographic, tumor, and clinical characteristics, middle-aged (adjusted odds ratio (aOR: 0.73, 95% CI, 0.70–0.75, *p* < 0.001) and elderly (aOR: 0.49, 95% CI, 0.47–0.50, *p* < 0.001) patients were less likely to present with metastatic disease as compared to the young cohort. In univariate analysis, the middle-aged cohort was less likely to present with high-grade cancer (OR: 0.79, 95% CI, 0.77–0.81, *p* < 0.001), but no differences in grade at presentation was noted between the young and elderly cohorts (*p* = 0.80). Patients who did not undergo surgery were more likely to have metastatic disease at presentation (aOR: 23.1, *p* < 0.001). The other independent predictors of metastatic disease of CRC at diagnosis are shown in Table 2. Patients in EraB had increased likelihood of presenting with metastatic disease on univariate analysis (OR = 1.03, 95% CI, 1.01–1.05, *p* = 0.001). In adjusted analysis, however, there was decreased likelihood of stage IV disease in EraB (aOR = 0.93, 95% CI, 0.91–0.95, *p* < 0.001).

### 2.4. Overall and CRC-Specific Survival

Of the 269,398 patients, 111,935 (41.6%) died over the study period of which 75,398 (67.4%) and 36,537 (32.6%) were in 2004–2009 and 2010–2015 respectively; *p* < 0.001. The median overall survival (OS) was not reached in young and middle-aged people over the study period but was 59 months for the elderly. However, OS was 121, 93, and 18 months for AJCC stages II, III, and IV, respectively. Median CRC-specific survival (CSS) was not reached in each age group. However, CSS by AJCC stage IV was 19 months. The 5-year OS for young individuals was 66.9% (95% CI, (66.3–67.5%)), middle aged, 67.2% (66.9–67.5%), and elderly, 49.5% (49.2–49.8%). In univariate analysis, middle-aged and elderly patients had inferior long-term survival. This persisted in an adjusted Cox proportional hazard model (aHR: 1.21, 95% CI, 1.19–1.24, *p* < 0.001 and aHR: 2.63, 95% CI, 2.57–2.70, *p* < 0.001). There was improved OS for all three age cohorts in EraB as compared to EraA (aHR: 0.96, 95% CI: 0.95–0.97, *p* < 0.001). The other independent predictors for survival are shown in Table 3.

The adjusted relative hazard (RH) of death as a function of age and as a continuous variable is shown in Figure 1. Overall survival was used to determine RH. We observed that there was evidence of a higher RH among younger individuals in the cohort. This seemingly plateaued until about 50 years at which point the RH increased dramatically. On stratified analysis by surgical intervention, the trend was reversed for younger patients but was similar for the rest of the age cohorts with an upward trend (Figure 2). For example, a 20-year-old patient who had surgical intervention has an RH of death of 3.8 (*p* < 0.001) compared to 15.2 (*p* < 0.001) for a 55-year-old patient. A similar trend was noted when stratified for the stage of CRC.

## 3. Discussion

Our results show that young individuals have a higher likelihood of presenting with metastatic CRC but have superior OS and CSS (Figure 3, Figure 4 and Figure 5). Although an increased incidence of CRC among young individuals has been all but established, the relationship between age and prognosis is not yet fully understood. Our results are consistent with a retrospective cohort study using SEER data collected between 1991 and 1999, which revealed that young patients (age 20 to 40 years) with CRC were more likely to present with later-stage and higher-grade tumors than their older (age 60 to 80 years) counterparts [8]. The authors showed that, despite presenting with more advanced cancers, overall five-year survival was only slightly lower in the young cohort when compared to the older cohort. Among those with stage IV cancer, 5-year survival was markedly longer in young versus older patients (18% and 6%, respectively).

Investigators hypothesized that young individuals with stage IV CRC demonstrated prolonged survival because they were more likely to be treated with adjuvant chemotherapy than older patients. Although there is no SEER data pertaining to specific chemotherapeutic regimens, young patients with stage IV CRC can often tolerate the toxic effects associated with FOLFOX-4 (oxaliplatin, folinic acid, and fluorouracil) and CLF-1 (irinotekan, folinic acid, and fluorouracil) that frequently require discontinuation of therapy in the elderly [9]. Manjelievskaia et al. evaluated data from the Central Cancer Registry of the United States Department of Defense and found that young (18 to 49 years) and middle-aged (50 to 64 years) patients were two to eight times more likely to receive systemic postoperative chemotherapy when compared to older patients (65 to 75 years) across all tumor stages [10]. However, there was no corresponding improvement in survival. Limitations of the study included a higher proportion of unknown chemotherapy regimens among the older group as compared to the younger age group as well as a small sample size for stratified analysis.

It is conceivable that evidence of superior OS and CSS in our study among young aged individuals was related to their ability to receive adjuvant and neoadjuvant chemotherapy. Young patients were also more likely than other age groups to have surgery in our data, which could have contributed to prolonged OS and CSS.

Recently, Chou et al. published a retrospective study evaluating the relationship between age and CRC prognosis [11]. Over 60,000 patients with CRC in Taiwan were divided into six age groups: age 40 or younger, 41 to 50 years, 51 to 60 years, 61 to 70 years, 71 to 80 years, and over 80 years. Individuals in the youngest age cohort had poorer overall survival than those in middle-aged patients (41 to 50 years and 61 to 70 years) and were more likely to present with tumors that exhibited aggressive histopathologic features. Investigators postulated that poor overall survival among young individuals with CRC is due to late detection. CRC screening before the age of 50 is uncommon and, thus, CRC is typically identified in young patients only after it has become symptomatic. However, despite a higher prevalence of aggressive tumors such as signet-ring cell carcinoma and mucinous adenocarcinoma among the youngest cohort, young individuals demonstrated superior overall survival as compared to those in the oldest cohorts (71 to 80 years and over 80 years). The authors attributed this finding to a higher rate of non-CRC-related death among those aged 71 or older. Limitations of that study include the absence of surgical treatment information and AJCC tumor staging data. Our data showed that both the middle-aged and elderly cohorts had poorer overall survival for all histologic CRC as well as shorter CSS for adenocarcinoma not otherwise specified and signet-ring cell carcinoma as compared to young patients.

To further examine these findings by Chou et al., we calculated RH of death as a function of age analyzed as a continuous variable (Figure 1). Using overall survival data, we calculated adjusted RH for increasing age. We observed that there was evidence of a higher RH for young individuals in the cohort, which seemingly plateaued until about 50 years of age at which point the RH increased dramatically. We then looked at RH stratified by surgical intervention and noted that this trend was reversed among younger patients but was similar for the rest of the age cohort with uptrend (Figure 2). Our analysis mirrors that of a 2014 study by Lieu et al. in which authors reviewed data on 20,023 patients with metastatic CRC and identified poorer OS among the youngest and oldest individuals who underwent treatment [12]. Investigators, therefore, concluded that younger and older patients might represent higher-risk populations. Our adjusted analysis for RH is consistent with the Lieu group. However, our data showed a relatively smaller RH among the young age group (Figure 1).

Our study contributes to the growing body of evidence indicating an increased risk of metastatic disease at presentation among young individuals. Those aged 20 to 49 years are significantly more likely than middle-aged and elderly individuals to have distant metastases at diagnosis (Figure 6) across the study period. However, young patients demonstrated the longest overall and CRC-specific survival among the three cohorts. Our findings are consistent with those published in a recent study by Rodriguez et al. who reviewed data from the Ontario Cancer Registry and found that patients under the age of 40 years exhibited improved overall survival as compared to older patients despite presenting with more advanced and aggressive disease [13].

There are several hypotheses to explain why young people are more likely to present with metastatic disease. First, CRC screening typically does not begin until the age of 50 years. While the American Cancer Society (ACS) guidelines were recently revised, the majority of practitioners have yet to implement earlier screening and, thus, late detection in the young is common. The United States Preventive Services Task Force (USPSTF), the American College of Physicians, and the American Society for Gastrointestinal Endoscopy have long recommended that individuals with an average risk for colorectal cancer (CRC) begin screening at the age of 50 [14,15,16], which most practitioners adhere to. In 2017, however, the American Cancer Society (ACS) published new guidelines suggesting that CRC screening start as early as the age of 45. The ACS cited a landmark study by Siegel et al. in which investigators had analyzed data from nearly half a million patients and noted a marked increase in the annual incidence of CRC in young individuals since the mid-1980s [17]. In their examination of CRC incidence trends in Surveillance, Epidemiology, and End Results (SEER) areas from 1974 to 2013, the Siegel group discovered that CRC incidence rates had consistently increased by 1% to 2.4% annually among adults age 20 to 39 years. A similar but slightly smaller increase in incidence was observed in those aged 40 to 54 years. Recently, Vuik et al. analyzed data collected over a 25-year period from 20 European national cancer registries to study trends in incidence rates of young adults with CRC. Investigators found that the incidence of colon cancer increased by 1.5% per year between 1990–2008 and by 7.4% annually between 2008–2016 [18]. Further reports confirm a similar incidence increase in Australia and New Zealand [19], China [20], Germany [21], and Pakistan [22].

Second, younger patients frequently delay seeking medical attention for CRC-associated symptoms [23]. Furthermore, young patients are more likely to tolerate chemotherapy regimens as compared to middle-aged and older adults. As described by Kneuertz et al., young patients are more likely to have a low Charlson-Deyo Comorbidity Index-a weighted score calculated based on the number of pre-existing comorbid conditions-when compared to individuals of a more advanced age [24]. These factors could have contributed to superior survival in this age category. Third, signet-ring cell carcinoma - an aggressive subtype of CRC that spreads rapidly and is characterized by late symptom manifestation - disproportionately affects young individuals [25]. Stratification of our data by histologic subtype revealed that young individuals had an increased risk of signet-ring cell carcinoma as compared to the older cohorts (Appendix A, Appendix A).

The mechanisms underlying superior survival among young individuals with CRC have yet to be fully elucidated. It is conceivable that intrinsic immunologic differences between young and old patients contributes to prolonged survival in the former group. Age-related immunosenescence, T-cell dysregulation, and systemic inflammation have been postulated to play a role in oncogenesis [26].

In addition to investigating the effect of age on CRC presentation and survival, we examined the relationship between race and CRC. Patients were stratified into six distinct ethnic groups: white (71.2%), African American (11.2%), Hispanic (9.2%), Asian (7.6%), Native American (0.6%), and unknown (0.2%). African Americans and Native Americans as compared to Hispanics were found to have an increased risk of metastatic disease at presentation and poorer survival, but OS was superior and likelihood of stage 4 disease was more common among whites and Asians. The association between race and CRC has been previously studied by Tammana and Laiyemo who identified several factors such as differences in diet and lifestyle, limited access to healthcare, and distinct tumor biology to account for the increased rate of advanced CRC and CRC-related mortality in some minority communities [27]. However, the most significant contributory factor may be poor patient education and insurance status. A 2015 study by May et al. concluded that racial minorities are less likely than whites to receive physician recommendations for CRC screening [28]. These findings suggest that establishing CRC screening guidelines by race is unlikely to have a significant effect on mortality in the absence of improved patient communication.

We also found that the chance of metastatic disease at presentation was lower in EraB (Appendix A**).** The recent decrease in metastatic disease warrants further investigation but may be attributed toward improved patient education and increased CRC screening. The percentage of adults who underwent CRC screening increased from 34.8% to 66.1% between 1987 and 2010 [29]. It appears likely that CRC is now more frequently detected at an early stage, which results in relatively lower number of patients presenting with advanced disease.

The increasing incidence of CRC in young individuals parallels that of obesity. As recently as 30 years ago, obesity was relatively rare among young and middle-aged adults. Today, however, approximately one-quarter to one-third of adults are affected [30]. Obesity has been identified as an independent risk factor for the development of CRC. Metabolic abnormalities, inappropriate insulin signaling, and activation of inflammatory pathways have been proposed as obesity-mediated mechanisms of oncogenesis [31]. Inadequate fiber intake is also common among overweight individuals and represents another obesity-associated risk factor for CRC [32]. Furthermore, a recent study by Hidayat et al. showed that body fatness at an early age increases the risk of CRC later in life [33]. Obesity in children and young adults is a relatively recent phenomenon. It is, therefore, possible that the increased incidence and presentation with distant metastasis of CRC in young people can be attributed in part to the rapid and unprecedented rise of obesity.

Our study has some limitations. First, we could not assess causation and subject it to selection bias. One would expect young patients to present with symptoms while older patients are diagnosed mostly through screening. Second, although multivariable analysis was used to control for potential confounders, there still remains the risk of residual confounding. Third, we did not have baseline data on comorbidities, which greatly influence treatment decision-making such as who undergoes surgeries. In addition, due to inaccurate reporting, SEER database has no information on non-surgical cancer directed therapies such as chemotherapy, which affects survival. Lastly, the SEER program collects data from population-based cancer registries, which covers about 28% of the USA population, and the results may not be generalizable to the entire USA population. However, this was minimized by adjusting for geographic regions. Despite these limitations, this study adds and provides a unique perspective on the stages of presentation and survival for CRC across age groups.

## 4. Materials and Methods

### 4.1. Study Design and Study Population

This is a retrospective cohort study using the SEER database for identification of CRC from all the registries captured in the SEER 18 program (San Francisco, Connecticut, Detroit, California, Kentucky, Louisiana, New Jersey, Greater Georgia, Hawaii, Iowa, New Mexico, Seattle, Utah, Alaska, San Jose-Monterey, Los Angeles, Rural Georgia, and Metropolitan Atlanta) who had a histologic diagnosis of colorectal cancers. SEER histology codes 8140, 8141, 8143, 8147, 8260, 8261, 8262, 8263, 8265, 8210, 8211, 8213, 8480, 8481, 8490, 8510, 8560, 8562, 8220, and 8221 were used for CRC diagnosed between 2004 and 2015. Patients were categorized into two groups based on the year of diagnosis: EraA (2004–2009) and EraB (2010–2015). An index registry was used to classify patients into various geographic regions: Midwestern (Detroit and Iowa), Western (California, Los Angeles, San Francisco, Hawaii, New Mexico, Seattle, Utah, Alaska, and San Jose-Monterey), Southern (Rural Georgia, Kentucky, Louisiana, Metropolitan Atlanta, and Greater Georgia), and North Eastern (New Jersey and Connecticut). The SEER registries continuously code and submit American Joint Committee on Cancer (AJCC) 6th and 7th Edition stages for all cancers diagnosed in 2010 and beyond. Patients diagnosed before 2010 are staged using the AJCC 6th edition only. The AJCC 6th edition was used in order to include all patients diagnosed between 2004 and 2015. Exclusion criteria included: (1) age younger than 20 years, (2) stage 0 or in situ tumor, (3) unknown tumor grade, (4) unknown tumor stage, (5) unknown site of primary tumor, (6) unavailable staging data, (7) patient deceased and cause of death unknown, and (8) history of previous cancer (Figure 7). This study did not require IRB approval since this is a publically available de-identified data and was granted access on the application request.

### 4.2. Data Source

The SEER database is comprised of data collected by the National Cancer Institute (https://www.cancer.gov). The SEER program collects and publishes cancer incidence and survival data using population-based cancer registries that include approximately 28% of the population of the United States. The program routinely collects data on patient demographics, tumor sites, tumor morphology, staging, surgical treatment, and follow-up.

### 4.3. Main Outcome Measures

Our primary outcome of interest was the likelihood of metastatic (stage IV) CRC at the time of diagnosis as compared to non-metastatic (stages I, II, or III) CRC. The secondary outcomes of interest were overall survival (OS) and CRC-specific survival (CSS). We estimated OS in months from the date of diagnosis to the date of death for non-survivors. The end of the follow-up period was used to ascertain OS for survivors. Patients were stratified into three groups based on age: young (20–49 years), middle-aged (50–69 years), and elderly (70 years or older).

### 4.4. Statistical Analysis

The baseline characteristics and group differences were compared by using Pearson’s Chi square (X^2^) test for proportions. Non-parametric variables were compared using the Mann Whitney-*U* test. The Kaplan-Meier method was used for survival analysis and the log rank test for equality of survival functions including assessing survival differences between patients diagnosed in EraA and EraB and among age groups. Continuous variables were analyzed with the student t-test. Stepwise multivariable Logistic and Cox regression models were built by using the forward method and adjusting for baseline demographics, treatment, and tumor characteristics. Variables included in the adjusted models had a *p*-value < 0.05 for the outcome of interest in the univariate analysis. These variables remained in the final model if they were still significant at *p* < 0.05 in the final adjusted model since a *p*-value < 0.05 was deemed statistically significant in this study. All statistical analyses were performed using Stata version 14.2 (StataCorp, College Station, TX, USA).

## 5. Conclusions

Young individuals with CRC have an increased likelihood of presenting with distant metastases as compared to the middle-aged and elderly. Yet, nevertheless, these individuals exhibit prolonged OS and CSS survival. The likelihood of metastatic disease at diagnosis for all age groups has been decreasing over time and has been persistently higher for young individuals over the study period. Screening for colorectal cancer is typically initiated at the age of 50. These findings support the new American Cancer Society (ACS) recommendations and suggest that earlier screening may improve colorectal cancer-associated morbidity and mortality.

## Figures and Tables

**Figure 1 cancers-10-00472-f001:**
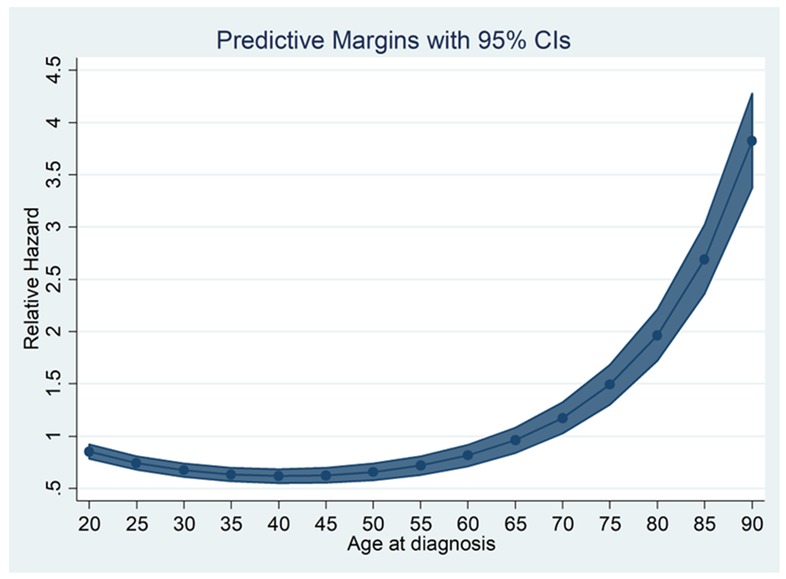
Relative Hazard (RH) of death as a nonlinear function of age at diagnosis. The RH is adjusted for other confounders. RH is high at a younger age and begins to decrease until age 50 at which point there is a gradual then sharp increase in the RH of death. RH was determined from overall survival (OS). CIs = Confidence intervals.

**Figure 2 cancers-10-00472-f002:**
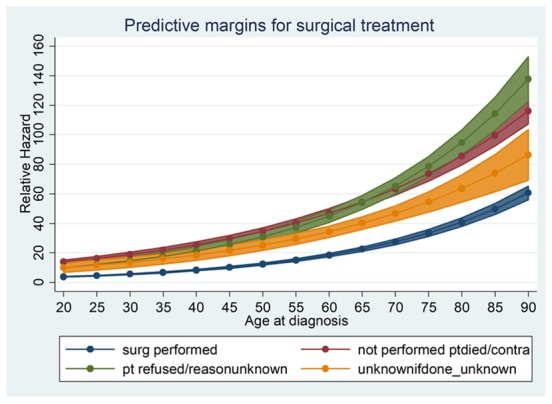
Relative Hazard (RH) of death as a nonlinear function of age at diagnosis, stratified for surgical intervention and non-surgical intervention. The RH is adjusted for other confounders. RH determined from overall survival (OS). Surg performed = those who had surgical intervention; Pt refused/reason unknown = patient refused or no reason given; not performed pt died/contra = no surgery performed or patient died; Unknownifdone_unknown = unknown if surgery was done.

**Figure 3 cancers-10-00472-f003:**
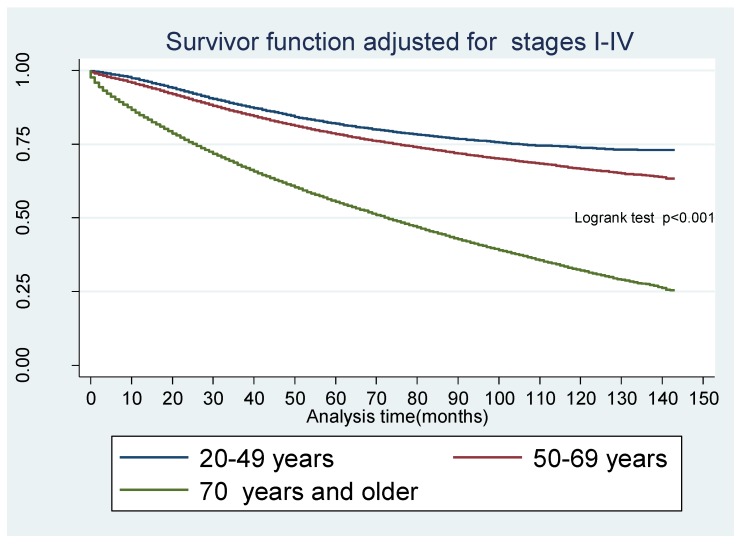
Kaplan Meier Survivor function for overall survival for age group adjusted for colorectal cancer stages. Colorectal cancer (CRC) stages and distribution by age group. Stage1, stage 2, stage 3, and stage 4 represent cancer stages at diagnosis. 20–49 = young age group 20 to 49, 50–69 = middle age group 50 to 69 and 70+ = 70 years and older. Stage 1 = stage I, Stage 2 = stage II, Stage 3 = stage III, Stage 4 = stage IV, I–IV = CRC stages I to IV.

**Figure 4 cancers-10-00472-f004:**
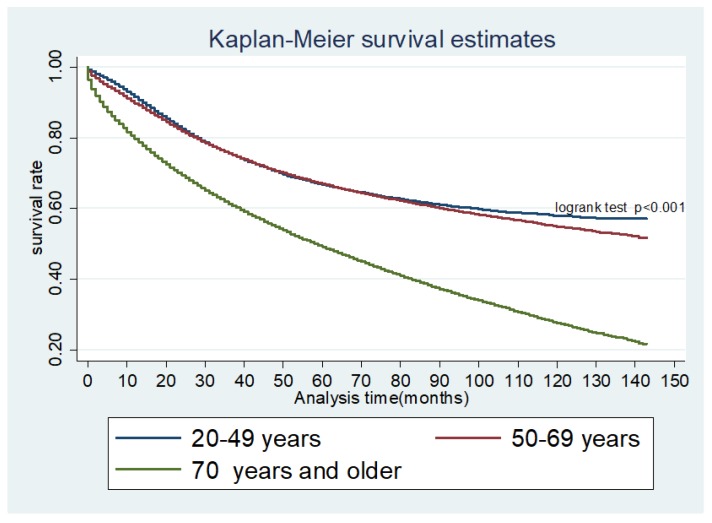
Kaplan Meier Survivor function: OS (overall survival) for age groups.

**Figure 5 cancers-10-00472-f005:**
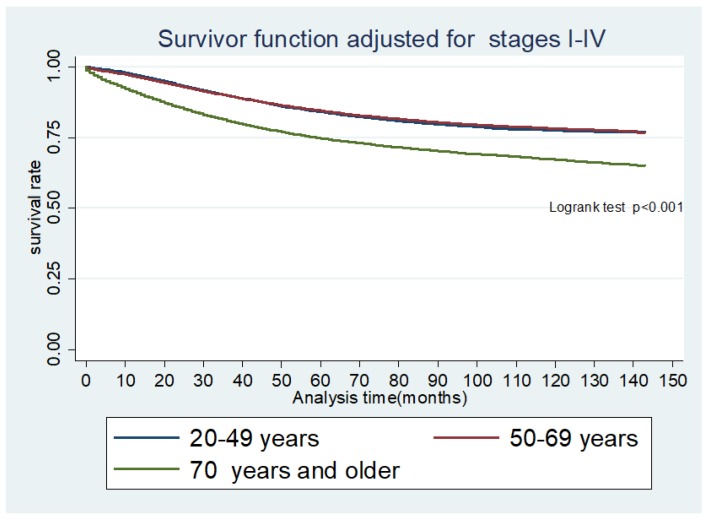
Kaplan Meier survivor function for Median CRC-specific survival (CSS) for an age group adjusted for CRC stages.

**Figure 6 cancers-10-00472-f006:**
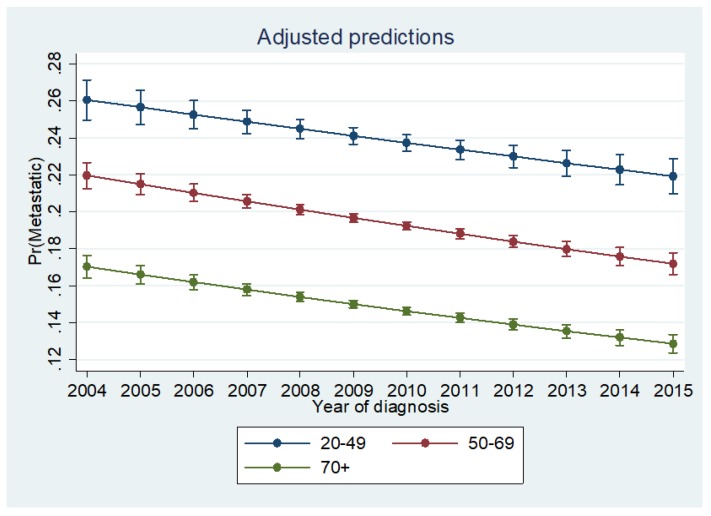
Predicted probabilities across age groups presenting with Stage IV disease for the follow-up period. 20–49 = age from 20 to 49, 50–69 = age from 50 to 69, 70+ = age 70 years and older; Pr (Metastatic) = probability of presenting with metastatic CRC.

**Figure 7 cancers-10-00472-f007:**
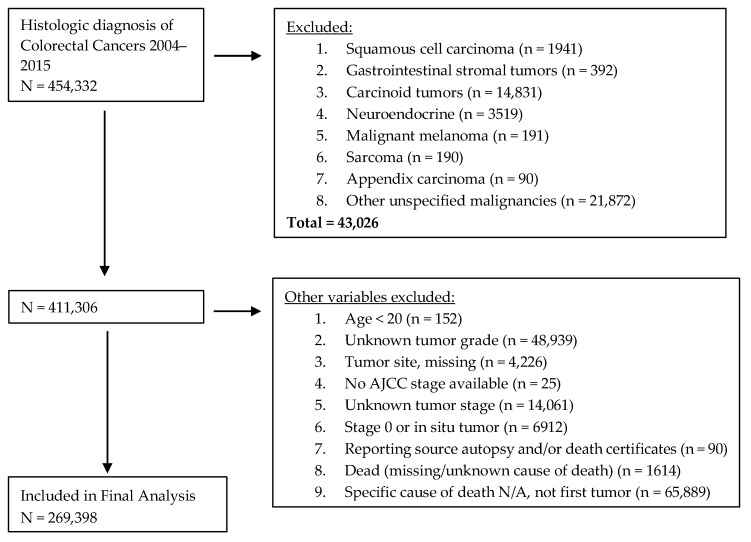
Patients’ cohort flow chart showing exclusion criteria.

**Table 1 cancers-10-00472-t001:** Demographic and tumor characteristic.

Variables	EraA (2004–2009) N = 138,650 (%)	EraB (2010–2015) N = 130,748 (%)	*p*-Value
**Age**			<0.001
20–49	15,502 (11.2)	16,166(12.4)	
50–69	60,288 (43.5)	62,620 (47.9)	
70+	62,860 (45.3)	51,962 (39.7)	
Age (SD)	66.8 ± 13.8	65.5 ± 13.8	<0.001
**Gender**			<0.001
Female	67,377 (48.6)	62,161 (47.5)	
Male	71,273 (51.4)	68,587 (52.5)	
**Race**			<0.001
Hispanic	12,691 (9.2)	15,430 (11.8)	
American Indian or Alaska Native	824 (0.6)	992 (0.8)	
Asian or Pacific Islanda	10,555 (7.6)	11,483 (8.8)	
Black	15,581 (11.2)	15,121 (11.6)	
Unknown	338 (0.2)	631 (0.5)	
White	98,661 (71.2)	87,091 (66.6)	
**Health Insurance**			<0.001
Uninsured	2193 (1.6)	4576 (3.6)	
Insured	57,776 (41.7)	106,695 (81.6)	
Medicaid	7320 (5.3)	16,993 (13.0)	
Unknown	71,361 (51.5)	2484 (1.9)	
**Marital Status**			<0.001
Married	76,867 (55.4)	68,775 (52.6)	
Divorced	12,204 (8.8)	12,529 (9.6)	
Separated/domestic partner	1269 (0.9)	1691 (1.3)	
Single	17,950 (13.0)	21,537 (16.5)	
Unknown	5087 (3.7)	7153 (5.5)	
Widowed	25,273 (18.2)	19,063 (14.6)	
**Geographic Region**			<0.001
Northeastern	23,289 (16.8)	15.54 (15.4)	
Midwestern	13,851 (10.0)	12,023 (9.2)	
Western	69,131 (49.9)	66,137 (50.6)	
Southern	32,379 (23.4)	32,273 (24.7)	
**Tumor Site**			<0.001
Right sided	49,460 (35.7)	46,018 (35.2)	
Left sided	52,435 (37.8)	47,508 (36.3)	
Transverse	9416 (6.8)	9042 (6.9)	
Rectum	25,355 (18.3)	26,164 (20.0)	
Large intestine NOS	1984 (1.4)	2016 (1.5)	
**Tumor Grade**			<0.001
Low	110,871 (80.0)	106,357 (81.4)	
High	27,779 (20.0)	24,391 (18.7)	
**Tumor Histology**			<0.001
Mucinous adenocarcinoma	12,134 (8.8)	9106 (7.0)	
Adenocarcinoma	124,564 (89.8)	119,334 (91.3)	
Signet ring cell adenocarcinoma	1367 (1.0)	1303 (1.0)	
Other (mixed, medullary, adenosquamous)	585 (0.4)	1005 (0.8)	
**Treatment by Surgery**			<0.001
Surgery performed	129,337 (93.3)	118,092 (90.3)	
No surgery	7709 (5.6)	10,699 (8.2)	
Patient Refused	1167 (0.8)	1206 (0.9)	
Unknown	437 (0.3)	751 (0.6)	
**AJCC Clinical Tumor Stage 6th Edition**			<0.001
I	34,181 (24.7)	30,909 (23.6)	
II	39,495 (28.5)	35,993 (27.5)	
III	40,319 (29.1)	39,942 (30.6)	
IV	24,655 (17.8)	23,904 (18.3)	

I–IV represent AJCC tumor stages, NOS = Non-Otherwise Specified, AJCC = American Joint Commission on Cancer.

**Table 2 cancers-10-00472-t002:** Independent predictors for metastatic disease at diagnosis.

Variables	Odds Ratio	95% Confidence Interval	*p*-Value
**Age**	Ref (20–49)		
50–69	0.73	0.70–0.75	<0.001
70 and older	0.49	0.47–0.50	<0.001
Age (continuous)			
**Male**	1.04	1.02–1.07	<0.001
**Race**	Ref (Hispanic)		
American Indian/Alaska Native	1.10	0.96–1.26	0.162
Asian or Pacific Islander	1.04	0.99–1.09	0.156
Black	1.26	1.20–1.32	<0.001
Unknown	0.22	0.16–0.29	<0.001
White	1.04	1.01–1.08	0.028
**Health Insurance**	Ref (Uninsured)		
Insured	0.78	0.73–0.83	<0.001
Medicaid	0.96	0.89–1.03	0.203
Unknown	0.86	0.80–0.92	<0.001
**Marital Status**	Ref (Married)		
Divorced	1.09	1.05–1.14	<0.001
Separated domestic partner	0.99	0.89–1.09	0.818
Single	1.06	1.02–1.09	0.001
Unknown	0.77	0.73–0.82	<0.001
Widowed	0.93	0.90–0.97	<0.001
EraA	Ref (EraA)		
**EraB**	0.93	0.91–0.95	<0.001
**Geographic Region (Northeastern)**	Ref (Northeastern)		
Midwestern	1.02	0.98–1.07	0.356
Western	1.02	0.99–1.05	0.237
Southern	1.05	1.02–1.09	0.005
**Tumor Site**	Ref (Right sided)		
Left sided	1.06	1.03–1.09	<0.001
Transverse	0.96	0.91–1.00	0.055
Rectum	0.50	0.48–0.52	<0.001
Large intestine NOS	1.63	1.51–1.77	<0.001
**Tumor Grade**	Ref (Low)		
High	2.12	2.07–2.18	<0.001
**Tumor Histology**	Ref (Mucinous adenocarcinoma)		
Adenocarcinoma	0.84	0.81–0.87	<0.001
Signet ring cell adenocarcinoma	1.13	1.02–1.24	0.019
Other (mixed, medullary, adenosquamous)	0.99	0.87–1.12	0.846
**Treatment by Surgery**	Ref (Surgery performed)		
No surgery	23.14	22.27–24.05	<0.001
Patient Refused	6.72	6.16–7.33	<0.001
Unknown	4.54	4.00–5.16	<0.001

Ref = reference.

**Table 3 cancers-10-00472-t003:** Independent predictors of survival.

Variables	Hazard Ratio	95% Confidence Interval	*p*-Value
**Age**	Ref (20–49)		
50–69	1.21	1.19–1.24	<0.001
70 and older	2.63	2.57–2.69	<0.001
**Male**	1.20	1.18–1.22	<0.001
**Race**	Ref (Hispanic)		
American Indian/Alaska Native	1.09	1.00–1.18	0.040
Asian or Pacific Islander	0.90	0.88–0.93	<0.001
Black	1.17	1.14–1.21	<0.001
Unknown	0.23	0.17–0.29	<0.001
White	1.04	1.02–1.07	<0.001
**Health Insurance**	Ref (Uninsured)		
Insured	0.78	0.75–0.82	<0.001
Medicaid	1.06	1.01–1.11	0.016
Unknown	0.86	0.82–0.89	<0.001
**Marital Status**	Ref (Married)		
Divorced	1.24	1.22–1.27	<0.001
Separated Domestic partner	1.26	1.19–1.34	<0.001
Single	1.33	1.30–1.35	<0.001
Unknown	1.09	1.05–1.12	<0.001
Widowed	1.46	1.43–1.49	<0.001
**Era (EraA)**	Ref (EraA)		
EraB (2010–2015)	0.96	0.95–0.98	<0.001
**Geographic Region**	Ref (Northeastern)		
Midwestern	1.04	1.02–1.07	<0.001
Western	1.03	1.01–1.05	0.002
Southern	1.12	1.10–1.14	<0.001
**Tumor Site**	Ref (Right sided)		
Left sided	0.91	0.90–0.92	<0.001
Transverse	0.99	0.97–1.02	0.543
Rectum	0.90	0.89–0.92	<0.001
Large intestine NOS	1.14	1.09–1.20	<0.001
**Tumor Grade**	Ref (low)		
High	1.44	1.42–1.46	<0.001
**Tumor Histology**	Ref (Mucinous adenocarcinoma)		
Adenocarcinoma	0.86	0.84–0.88	<0.001
Signet ring cell adenocarcinoma	1.34	1.27–1.41	<0.001
Other (mixed, medullary, adenosquamous)	1.12	1.04–1.20	0.004
**Treatment by Surgery**	Ref (Surgery performed)		
No surgery	2.42	2.37–2.47	<0.001
Patient refused	2.54	2.41–2.67	<0.001
Unknown	1.77	1.63–1.92	<0.001
**AJCC Clinical Tumor Stage 6th Edition**	Ref (stage 1–3)		
Stage IV	5.07	5.00–5.15	<0.001

Ref = reference.

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
