# Peer review of "Colorectal Cancer Presentation and Survival in Young Individuals: A Retrospective Cohort Study"

_cancers, 2018, doi:10.3390/cancers10120472_

Round 1
Reviewer 1 Report
The manuscript is well written, well-organised, clear and concise. The figures are well presented. The paper was a pleasure to review. The authors give a detailed analysis of the outcomes for young adults with colorectal cancer. Though the findings are not novel, the study covers a large population, and the findings are of interest to a broad sector of the medical and scientific community. The limitations have been clearly stated. The only comment I have is that in Table 1, some of the variables which have been analysed en block by Chi-squared analysis in which proportions go up or down across EraA and EraB may have obscured the granularity of the data.
Author Response
The manuscript is well written, well-organised, clear and concise. The figures are well presented. The paper was a pleasure to review. The authors give a detailed analysis of the outcomes for young adults with colorectal cancer. Though the findings are not novel, the study covers a large population, and the findings are of interest to a broad sector of the medical and scientific community. The limitations have been clearly stated.
Thank you for the kind words. We sincerely appreciate your favorable review of our mansucript.
The only comment I have is that in Table 1, some of the variables which have been analysed en block by Chi-squared analysis in which proportions go up or down across EraA and EraB may have obscured the granularity of the data.
We appreciate the thoughtful suggestion and agree that the formatting of the table may obscure the granularity of the data. However, we feel that it was critical to stratify the subjects into EraA and EraB; by doing so, variables which have more than two sub-variables (for example, “Geographic Region”) could not be analyzed individually. If we were to analyze the data across EraA and EraB, important information pertaining to the changing proportions of CRC in recent years for each era, would be lost.

Reviewer 2 Report
I read with deep interest the manuscript entitled “Colorectal Cancer Presentation and Survival in Young Individuals: A Retrospective Cohort Study” provided by Ulanja and coauthors. The article is overall well written and focuses on a currently clinical unmet need such as colorectal cancer (CRC) among patients younger than 50 years of age. The authors presented retrospective data obtained analyzing SEER 18 registry data and they documented increase of stage IV CRC at diagnosis. However, from their analysis, young individuals experienced a better adjusted survival.
Two are the major points:
The first observation is about the real impact of this manuscript to the already available literature. Even if CRC among young individuals is a currently hot topic of epidemiologic and clinical research as well, it is not clear which are the original data provided by this manuscript if compared to the already available articles in the literature. Indeed, various articles already available in the literature analyzed stage at presentation as well as prognosis of early-onset CRC retrieving data from SEER database even if considering different years ranges (O’Connell et al., 2004; Sultan et al., 2010; Wang et al., 2015; Abdelsattar et al., 2016; Li et al., 2014; Hawk et al., 2014). This manuscript presents a further cohort of CRC obtained from a different years range but it does not solve the debate on prognosis of young CRC.
The second point is about age group subdivision. Age groups subdivision carried out by the authors for this manuscript is arbitrary, as well as in the majority of the already published articles on early-onset CRC. This subdivision derived from screening age cut-off (50 years of age) might shadows different biology between young and late onset CRC. To overcome this limit, it would be of greater interest to analyze this cohort of patients considering age as a continuous variable as presented by Lieu et al. JCO, 2014. This analysis could potentially add original data and relevance to the manuscript since this SEER age analysis is still lacking from the literature.
Other minor points are:
1. The term “results” (page 1 line 36) is representing prognosis or survival or other?
2. Considering that data from SEER database have been already analyzed it would be better to report the years range even in the introduction at page 1 line 38.
3. Recently it has been presented an abstract at the European Gastroenterology Week describing an increase incidence of CRC among the younger even in the European Union (Vuik et al, 2018 - abstract). In addition to that, at ESMO 2018 another abstract described an increase incidence of CRC among the younger in the United Kingdom (Exarchakou et al, 2018 – Abstract - Ann Oncol 2018 suppl_8).
4. In the discussion section, when dealing with factors conditioning a better overall survival of young CRC if compared to middle-aged and elderly despite later and more aggressive features at diagnosis, the authors should discuss also about clinical aspects such as a likely better tolerance of chemotherapy regimens by the younger and about the more usual absence of comorbidities (Charlson Comorbidity Index 0) among the younger (Kneuertz et al., 2015). Both these aspects can condition the outcome to current surgical and medical treatments.
Author Response
I read with deep interest the manuscript entitled “Colorectal Cancer Presentation and Survival in Young Individuals: A Retrospective Cohort Study” provided by Ulanja and coauthors. The article is overall well written and focuses on a currently clinical unmet need such as colorectal cancer (CRC) among patients younger than 50 years of age. The authors presented retrospective data obtained analyzing SEER 18 registry data and they documented increase of stage IV CRC at diagnosis. However, from their analysis, young individuals experienced a better adjusted survival.
Two are the major points:
The first observation is about the real impact of this manuscript to the already available literature. Even if CRC among young individuals is a currently hot topic of epidemiologic and clinical research as well, it is not clear which are the original data provided by this manuscript if compared to the already available articles in the literature. Indeed, various articles already available in the literature analyzed stage at presentation as well as prognosis of early-onset CRC retrieving data from SEER database even if considering different years ranges (O’Connell et al., 2004; Sultan et al., 2010; Wang et al., 2015; Abdelsattar et al., 2016; Li et al., 2014; Hawk et al., 2014). This manuscript presents a further cohort of CRC obtained from a different years range but it does not solve the debate on prognosis of young CRC.
Thank you for the comments. We agree that our paper does not conclusively solve the debate on prognosis of young CRC. However, we believe that the data contributes to the growing body of evidence
The second point is about age group subdivision. Age groups subdivision carried out by the authors for this manuscript is arbitrary, as well as in the majority of the already published articles on early-onset CRC. This subdivision derived from screening age cut-off (50 years of age) might shadows different biology between young and late onset CRC. To overcome this limit, it would be of greater interest to analyze this cohort of patients considering age as a continuous variable as presented by Lieu et al. JCO, 2014. This analysis could potentially add original data and relevance to the manuscript since this SEER age analysis is still lacking from the literature.
We appreciate the astute observation. We have read the excellent paper by Lieu et al. and agree that analyzing the cohort of patients considering age as a continuous variable significantly improves the applicability of the data. We have revised the manuscript accordingly (Introduction: Lines 88-94; Discussion: Lines 175-186; Figure 1; Figure 2).
Other minor points are:
1. The term “results” (page 1 line 36) is representing prognosis or survival or other?
Thank you for the suggestion. We concur and have clarified this point.
2. Considering that data from SEER database have been already analyzed it would be better to report the years range even in the introduction at page 1 line 38.
We agree with the recommendation and have revised the introduction to report the age range (Introduction: Line 42).
3. Recently it has been presented an abstract at the European Gastroenterology Week describing an increase incidence of CRC among the younger even in the European Union (Vuik et al, 2018 - abstract). In addition to that, at ESMO 2018 another abstract described an increase incidence of CRC among the younger in the United Kingdom (Exarchakou et al, 2018 – Abstract - Ann Oncol 2018 suppl_8).
Thank you for the excellent recommendations. We have reviewed the above-listed papers and agree that the authors’ findings are pertinent to our study. A discussion of the Vuik et al. abstract can now be found in in the Discussion, Lines 213-216.
4. In the discussion section, when dealing with factors conditioning a better overall survival of young CRC if compared to middle-aged and elderly despite later and more aggressive features at diagnosis, the authors should discuss also about clinical aspects such as a likely better tolerance of chemotherapy regimens by the younger and about the more usual absence of comorbidities (Charlson Comorbidity Index 0) among the younger (Kneuertz et al., 2015). Both these aspects can condition the outcome to current surgical and medical treatments.
We appreciate the suggestion. The data from the Kneuertz et al. study is important and relevant to our findings. We have therefore included a brief discussion of the Kneurtz group paper and the Charlson-Deyo Comorbidity Index in the Discussion, Lines 219-223.
